# In vivo CRISPR/Cas9 knockout screen: TCEAL1 silencing enhances docetaxel efficacy in prostate cancer

Linda K Rushworth[1,2,*], Victoria Harle[1,2,*], Peter Repiscak[1,2,*], William Clark[2], Robin Shaw[2], Holly Hall[2] (iD), Martin Bushell[1,2], Hing Y Leung[1,2,†] (iD), Rachana Patel[2,†]

**Docetaxel chemotherapy in metastatic prostate cancer offers only a modest survival benefit because of emerging resistance. To identify candidate therapeutic gene targets, we applied a murine prostate cancer orthograft model that recapitulates clinical invasive prostate cancer in a genome-wide CRISPR/Cas9 screen under docetaxel treatment pressure. We identified 17 candidate genes whose suppression may enhance the efficacy of docetaxel, with transcription elongation factor A–like 1 (*Tceal1*) as the top candidate. TCEAL1 function is not fully characterised; it may modulate transcription in a promoter dependent fashion. Suppressed TCEAL1 expression in multiple human prostate cancer cell lines enhanced therapeutic response to docetaxel. Based on gene set enrichment analysis from transcriptomic data and flow cytometry, we confirmed that loss of TCEAL1 in combination with docetaxel leads to an altered cell cycle profile compared with docetaxel alone, with increased subG1 cell death and increased polyploidy. Here, we report the first in vivo genome-wide treatment sensitisation CRISPR screen in prostate cancer, and present proof of concept data on *TCEAL1* as a candidate for a combinational strategy with the use of docetaxel.**

## Introduction

Prostate cancer is the second most common cause of cancer deaths in men in the Western world (1). Androgen deprivation therapy (ADT) remains the first-line hormonal treatment option, whereas docetaxel is currently the standard chemotherapy drug routinely used to treat metastatic prostate cancer. Treatment with docetaxel, however, leads to only a modest increase in median survival of 10 mo (2). A second-line chemotherapy drug, cabazitaxel, has been approved. Similarly to docetaxel, cabazitaxel only offers a modest survival benefit of just 2.4 mo (3). Recent evidence from clinical trials giving hormone-sensitive patients upfront treatment of

docetaxel in combination with ADT has demonstrated a robust increase in survival. Subsequently, upfront ADT combination therapy with either chemotherapy or androgen receptor (AR) pathway inhibitors has become routinely used (2, 4, 5, 6). Despite an initial docetaxel response, most tumours relapse within 2–3 yr via resistance mechanisms, either de novo or by acquired treatment resistance (7). Thus, there is an unmet need for additional combination approaches to improve the efficacy of docetaxel.

The CRISPR/Cas9 system consists of two components, a Cas9 endonuclease and a single-stranded guide RNA (sgRNA), and is a powerful genome-editing tool. Cas9 can be directed to a specific gene locus by a sgRNA which is matched to targeted genomic loci, leading to double strand breaks and subsequent indels potentially resulting in loss of gene function (8). CRISPR-based screening represents a powerful tool for studying biological processes, including those involved in cancer (9, 10). Targeted CRISPR screens have been used in cancer studies, and more recently, genome wide screens begin to comprehensively identify genes required for a phenotype of interest. In vivo screens are preferred over in vitro screens, with in vivo models mimicking human disease better and the incorporation of tumour microenvironment in the model (11). However, in vivo CRISPR screens are significantly more demanding to perform, and none have been reported for prostate cancer. CRISPR screens can provide a wealth of information, as genes that are potentially involved in the treatment or process of interest can be identified by comparing the abundance of individual sgRNAs. Negatively selected sgRNAs signify that the target gene may be required for cellular survival and/or proliferation under the screening conditions.

To our knowledge, we conducted the first in vivo dropout docetaxel sensitisation CRISPR screen in prostate cancer. Using a whole genome approach, we transduced Cas9-expressing murine prostate cancer cells (from a *Probasin-Cre Pten^{fl/fl} Spry2^{fl/+}* tumour) (12, 13) with a whole genome library. Mice injected with these cells were treated with docetaxel or vehicle, and the resulting tumours were deep sequenced to profile the abundance of individual gRNA species. In a drop-out screen, we focussed on negatively selected

---

[1]Institute of Cancer Sciences, College of Medical, Veterinary and Life Sciences, University of Glasgow, Glasgow, UK   [2]Cancer Research UK Beatson Institute, Glasgow, UK

Correspondence: h.leung@beatson.gla.ac.uk
*Linda K Rushworth, Victoria Harle, and Peter Repiscak contributed equally to this work
†Hing Y Leung and Rachana Patel contributed equally to this work

genes, which may signify a potential role for cells to survive docetaxel treatment. We successfully validated the top target *TCEAL1* in both murine and human prostate cancer cells. We further identified cell cycle alterations to be associated with enhanced treatment response upon combined TCEAL1 silencing and docetaxel treatment.

# Results

## Establishing an orthograft model for in vivo CRISPR screening

Inactivation of tumour suppressors such as PTEN and Sprouty2 (SPRY2) drives aggressive treatment resistant prostate cancer (12). Genomic alterations of *SPRY2* and *PTEN* as part of the RAS/ERK and PI3K/AKT pathways, respectively, have been detected in ~40% of metastatic prostate cancer patients (SU2C/PCF Dream Team) (14) (Fig 1A). The genetically engineered mouse model with Probasin-mediated deletion of *Pten and Spry2* (namely *PbCre Pten^{fl/fl} Spry2^{fl/+}*, referred to as the SP model hereafter) models clinical invasive prostate cancer (12, 13). Of note, tumours from the SP model have an adenocarcinoma phenotype with evidence of glandular differentiation, thus recapitulating the most common type of clinical prostate cancer. Prostate tumour weights were higher in the SP mice than those with *Pten* deletion alone (Fig 1B), suggesting that combined altered RAS/ERK and PI3K/AKT signalling promotes prostate tumorigenesis. We then generated and characterised a murine prostate cancer cell line from an SP tumour, hereafter referred to as SP1 cells (Fig S1A). SP1 cells have been used in previous studies (12, 15), and importantly for clinical relevance, they express AR (Fig S1B).

Orthotopic injection of SP1 cells results in reproducible formation of prostate tumours. In an optimisation experiment, mice bearing SP1 tumours were treated with docetaxel (6 mg/kg at 4 d intervals) (16). Docetaxel treatment significantly extended the survival of experimental mice (Fig 1C, median survival extending from 33 to 38 d), with reduced tumoral Ki67 staining (Figs 1D and S1C). Despite initial response to chemotherapy, all of the mice demonstrated persistent tumour growth and no mice survived beyond 40 d. Thus, SP1-derived orthografts represent a clinically relevant model for an in vivo CRISPR/Cas9 screen to identify novel genes/pathways that influence tumour response to docetaxel (Fig 1E).

## In vivo whole genome CRISPR/Cas9 screen in a prostate cancer orthograft model

SP1 cells were transfected with Cas9-EGFP (Fig 1F) and subjected to a double live cell sort to collect EGFP expressing cells. As sgRNAs require nuclear nuclease activity of Cas9, nuclear Cas9-EGFP expression was confirmed (Fig 1G). SP1-Cas9 cells were transduced with the CRISPR library (GeCKOv2 library A; Addgene) and $10^7$ cells were injected into one of the anterior prostates of each CD-1 nude mouse. Mice were randomised for vehicle (n = 9) or docetaxel (n = 5) treatment. Figs 1H and S1D show the full workflow of the screen. At the end of the screen, docetaxel treated and control mice had comparable tumours (Fig 1I). Tumour samples were deep sequenced, along with control samples (including GeCKO plasmid A

input library and cells transduced with the sgRNA library) to confirm library representation before injection and treatment.

## Bioinformatic analysis identifies negatively selected genes

The average number of mapped reads across conditions was 15 million, with a minimum of 4.5 million; even the minimum depth will provide sufficient theoretical coverage of more than 68 reads per sgRNA (Table S1). The representation of the sgRNA library is shown as a boxplot distribution in Fig 2A. Distribution of unique sgRNA abundances across different conditions were further examined by plotting cumulative probability distributions as a function of normalised reads (Fig 2B). The plasmid and transduced SP1-Cas9 cells (before injection) had excellent sgRNA distribution, with detected sgRNAs representing >98% of total sgRNA. Whereas sgRNA for essential survival genes were anticipated to be under-represented in the pre-injection transduced SP1-Cas9 cells relative to the plasmid, the sgRNA representation in the two groups correlated significantly (Pearson, r = 0.94) (Fig S2A), which suggested suboptimal performance of the screen resulting in the risk of false negatives. Nonetheless, analysis of the prostate tumours (both vehicle and docetaxel treated) confirmed some loss in the amount of detected sgRNAs (Table S1), with an average of 83% of genes being represented in the library across all samples. As expected, the plasmid and transduced SP1-Cas9 cell samples cluster away from the prostate tumours, and tumours cluster by treatment (vehicle or docetaxel) (Fig 2C).

Genes can be identified by comparing the abundance of individual sgRNAs that are positively or negatively enriched in the cell population compared with control tumours in vehicle treated mice. Waterfall plots with ranked sgRNA abundance were prepared (Fig S2B), with individual gRNAs for gene hits of interest highlighted. Comparing tumours in vehicle and docetaxel treated mice, we identified 17 candidate negatively selected genes after chemotherapy (padj < 0.25; Figs 2D and S2C and Table 1), including 15 coding genes (eight with human orthologues) and two microRNAs.

## *Tceal1* is identified as the top candidate among the negatively selected genes

From the 17 highlighted genes, six genes had highly significant adjusted *P*-values at <0.05, among which two genes have human orthologues (*TCEAL1* and *CUL9*) (Table 1). The most significant negatively selected gene in the screen was *Tceal1* (transcription elongation factor A−like 1) ($\log_2$ fold change = −3.4; padj = 0.0267), and the possibility of off-target hits for *Tceal1* sgRNAs was excluded (Table S2). *TCEAL1* is part of a gene family of transcription elongation factor A−like proteins, which includes TCEAL1 − 9, clustered on the X chromosome (Xq22.1-2). TCEAL1 is hypothesised to modulate transcription both positively and negatively depending on the target promoters (17). In our screen with orthograft bearing mice, we identified 19 metastatic lesions (six in epididymal fat and four in bladder in the vehicle group mice, five in epididymal fat, and four in bladder in the docetaxel treated mice) for analysis. Of note, *Tceal1* was also implicated to be a significantly dropped-out gene in metastases following docetaxel treatment ($\log_2$ fold change = −3.3 *P* = 0.0006; false discovery rate [FDR] 0.0082).

We identified *Cul9* as another top hit (log$_2$ fold change = −4.3; padj = 0.0364). CUL9 is part of a complex that mediates ubiquitination and degradation of survivin and is required to maintain microtubule dynamics (18). CUL9 interacts with paclitaxel to regulate microtubule stability (18), thus confirming the validity of hits from our screen. With pdj < 0.25, *WDR72* (WD Repeat domain 72) is one of the six genes with human orthologues (Table 1) and is underrepresented at −2.2 log$_2$ fold (padj = 0.1848). Mutations in *WDR72* are associated with amelogenesis imperfecta hypomaturation type 2A3 (19, 20), and altered *WDR72* expression has been reported in lung cancer stem cells (21). In the presence of docetaxel, silencing of *Tceal1*, *Cul9*, or *Wdr72* expression in SP1 cells resulted in significant additional reduction of cell numbers relative to each treatment alone (Figs 2E and S3A and B). Similarly, siRNA-mediated knockdown of the three genes enhanced the response to docetaxel in human PC3M prostate cancer cells (Figs 2E and S3A and C).

Focussing on *Tceal1* as the top hit, LNCaP, DU145, and CWR22 human prostate cancer cell lines were also sensitised to docetaxel treatment upon suppressed *TCEAL1* expression (Figs 2E and S3A). Although all of the four human PCa cell lines express easily detectable levels of TCEAL1 protein, the benign prostate epithelial RWPE cells have almost undetectable levels of TCEAL1 protein expression (Fig S4A). It is worth noting that RWPE cells do express TCEAL1 mRNA at an easily detectable level (Fig S4B). Besides pooled siRNA, two individual *TCEAL1* siRNAs were confirmed to suppress *TCEAL1* expression and reduce proliferation in PC3M cells (Figs 3A and B and S4C). Interestingly, siRNA-mediated silencing of *TCEAL1* mRNA expression did not sensitise RWPE cells to docetaxel treatment (Figs 3C and S4B), perhaps because of the fact that RWPE cells have very low levels of TCEAL1 protein expression. For the first time, TCEAL1 is implicated in enhancing docetaxel anti-cancer effects in prostate cancer.

### Cell cycle profile analysis after suppression of TCEAL1 expression

We next studied the cell cycle profile of synchronised PC3M cells by flow cytometry. In isolation, docetaxel (2 nM for 48 h) significantly suppressed G1 and up-regulated G2M and S phase subpopulations (Figs 3D and E and S4D and Table S3). TCEAL1 knockdown alone resulted in more modest changes; however, there was a small but significant decrease in G1 and increase in polyploidy. With combined *TCEAL1* siRNA and docetaxel treatment, the cell cycle profile was altered compared with each treatment alone, with all stages of the cell cycle (except the S phase) being significantly changed. Interestingly, the percentages of both sub-G1 and polyploid cells were significantly increased, possibly because of aberrant mitosis leading to altered DNA content. The percentage of G2M cells was decreased with combined treatment compared with docetaxel alone (Figs 3D and E and S4D and Table S3). Taken together, these data suggest that the combined treatment altered the cell cycle in a manner distinct from the individual treatments.

### Transcriptomic analysis of PC3M cells with suppressed TCEAL1 expression

To gain further insight into TCEAL1-mediated functions, and how TCEAL1 influences cancer response to docetaxel treatment, RNA sequencing was conducted using samples prepared from PC3M cells after TCEAL1 knockdown with/without docetaxel treatment (2 nM for 48 h). TCEAL1 knockdown accounted for most of the differences in gene expression as seen in the principal component analysis, whereas docetaxel treatment had a lesser effect (Fig S4E). We analysed the transcriptome upon TCEAL1 knockdown in the first instance. Genes that were up-regulated included multiple biological processes related to cell cycle and DNA replication (Fig 3F, highlighted in red), whereas down-regulated genes were generally related to translation (Fig 3G). TCEAL1 expression was potently suppressed by *TCEAL1* siRNA treatment which has only minor effects on the expression of other TCEAL genes (Fig S4F).

5,169 genes were significantly altered after combined TCEAL1 siRNA and docetaxel treatment, with only 623 for docetaxel and 2,960 for TCEAL1 knockdown alone (fold change > 1.5, *P*.adj < 0.05) (Figs 4A and S5A). Almost half (n = 2,538) of the differentially expressed genes upon combined TCEAL1 loss and docetaxel treatment were unique and not observed after single treatment (Fig 4A). Based on the Hallmark gene sets for defined biological states and processes, the gene expression data in docetaxel treated cells revealed multiple up-regulated gene sets with positive normalised enrichment scores (Fig 4B). In contrast, cells with TCEAL1 loss alone tend to have negatively enriched gene sets. Some of the gene sets that were positively enriched by docetaxel were negatively enriched by TCEAL1 alone (e.g., KRAS signalling up, myogenesis, and epithelial mesenchymal transition) with combined treatment showing no enrichment, suggesting mutual compensation, whereas enrichment of other gene sets were common to all three treatments (e.g., mitotic spindle, oxidative phosphorylation, and myc targets v1 and v2) (Fig 4B).

Gene sets for G2M checkpoint and E2F targets were enriched with TCEAL1 loss (NES 1.31, padj 0.0913; NES 1.50, padj 0.0128 respectively), and further enriched with the addition of docetaxel (NES 1.53, padj 0.0094; NES 1.67, padj 0.0014 respectively), suggesting that functional effects of combined TCEAL1 loss and docetaxel may be related to the cell cycle (Fig 4B and C), in line with our flow cytometry data on TCEAL1-mediated effects. Focussing on expression of E2F target genes, combined TCEAL1 loss and docetaxel treatment has the greatest effects, whereas TCEAL1 loss alone resulted in smaller but significant effects (Fig 4C). E2F transcription factors transcriptionally control genes involved in the cell cycle and DNA replication. We selected some of the genes with the most altered expression levels upon TCEAL1 loss and validated the findings in independent PC3M cell cultures (Figs 4D and S5B). The genes included those involved in cell cycle checkpoints, such as *CHEK1* and *CDC25A*, as well as those involved in the separation of chromatids during mitosis. Interestingly, some E2F family members were themselves up-regulated upon TCEAL1 knockdown (Fig S5B). In summary, TCEAL1 was identified in an in vivo CRISPR/Cas9 screen to enhance the effect of docetaxel, with associated changes in cell cycle profile and E2F target expression.

## Discussion

The use of taxanes is well established in metastatic prostate cancer, but the survival benefits from docetaxel and cabazitaxel chemotherapy are modest (2, 3). There is therefore an urgent unmet requirement

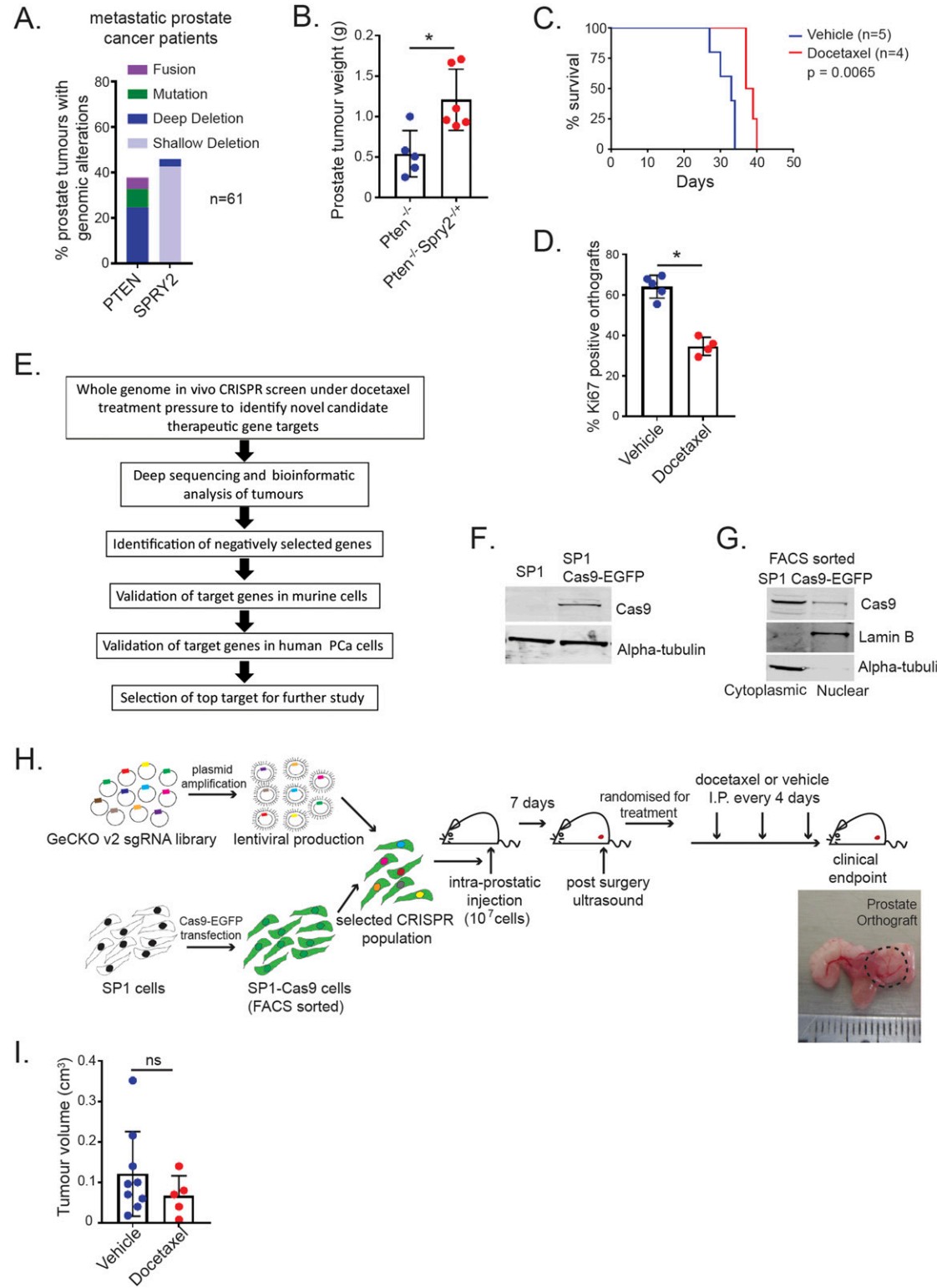

**Figure 1. In vivo whole genome CRISPR/Cas9 screen.**
**(A)** *PTEN* and *SPRY2* genomic alterations in metastatic prostate cancer patients with taxane treatment (SU2C/PCF Dream Team, 2015). **(B)** Non-cystic prostate tumour weights from indicated mice at clinical end point (*Pten$^{-/-}$*, n = 5; *Pten$^{-/-}$ Spry2$^{-/+}$*, n = 6; *P < 0.05; Mann–Whitney test; mean values ± SD are shown). **(C)** Kaplan–Meier plot for overall survival of SP1 prostate orthograft bearing mice treated as indicated (log-rank Mantel–Cox test). **(D)** IHC quantification of Ki67 staining in SP1 prostate tumour orthograft sections from CD-1 nude immunocompromised mice treated as indicated (vehicle, n = 5; docetaxel, n = 4; *P < 0.05; Mann–Whitney test; mean values ± SD are shown). **(E)** Schematic of the workflow of the CRISPR drop-out screen, bioinformatics analysis and target validation. **(F)** Western blot images to confirm expression of Cas9

for improved therapies. We applied the first in vivo prostate cancer whole genome CRISPR screening to study drug sensitisation to identify genes and pathways that sensitise prostate cancer cells to docetaxel treatment using a clinically relevant orthotopic mouse model.

We injected SP1 cells orthotopically into immunocompromised CD-1 mice to recapitulate the prostate cancer microenvironment, although they did lack a normal adaptive immune system. We applied the two-vector murine CRISPR knockout GeCKOv2 pooled library to provide genome wide coverage. The in vivo experimental design was developed within the limit of SP1 cell number we could inject per mouse. Managing the cell number restriction, Library A alone from the GeCKOv2 library (number sgRNA = 67,405) was selected, targeting the entire genome along with all relevant controls and achieving a library representation (cells per lentiviral CRISPR construct) at 100-fold. Under-representation (or dropout) of sgRNA for specific genes upon docetaxel treatment suggests the inability of cells to survive when the implicated sgRNAs are present which are expected to suppress the expression of the target genes. Hence, the target genes of "drop-out" sgRNA signifies genes required for cells to resist docetaxel.

Comparison of sgRNA in the pre-injection-transduced SP1-Cas9 cells to the plasmids was expected to highlight under-representation of sgRNA for essential genes. The inability of our screen to confirm this under-representation is likely a consequence of inadequate coverage of sgRNA for individual target genes. Cas9 expression in SP1-Cas9 cells appears to be satisfactory, as determined by Western blot (Fig 1F and G); however, the screen could also have been affected by suboptimal Cas9 activity (9). Instead of restricting the screen to library A, a more focussed screen with the inclusion of more sgRNA per gene may provide better coverage (22, 23) and avoid the risk of false negatives in the screen. Nonetheless, the screen identified 17 genes with negatively selected sgRNAs, including two miRNAs, in orthografts upon docetaxel treatment, signifying potential novel therapeutic targets. We validated *Tceal1*, *Cul9*, and *Wdr72* in the murine SP1 cells. CUL9 has previously been described as having a combination effect with paclitaxel, and loss of WDR72 only sensitised two of the cell lines tested to docetaxel. *Tceal1* was the gene with the most significant negatively selected sgRNAs, sensitising all prostate cancer cell lines tested to docetaxel, and given its putative role in transcription, we chose *TCEAL1* as our top target for further study. In addition, the finding of *TCEAL1* dropout in both prostate orthografts and associated metastasis after docetaxel treatment highlighted the importance of TCEAL1 in in vivo prostate carcinogenesis.

TCEAL1 was identified as a phosphoprotein similar to transcription factor SII (24, 25) that can modulate promoter function. Importantly, TCEAL1 can either repress promoter function or up-regulate transcriptional activity in a context-dependent manner (17). *TCEAL1* is part of a family of transcription elongation factor A–like genes. *TCEAL* family members have not been widely studied, and the studies that have been published describe varying roles for these genes in cancer. Whereas TCEAL2 up-regulation was reported to associate with poor prognosis for serous ovarian cancer patients (26), TCEAL-1, 4, and 7 were reported to be down-regulated in different tumour types (27, 28, 29). To date, the expression of TCEAL has not been implicated in tumour response to treatment.

Gene set enrichment analyses (GSEA) of RNA sequencing showed that several pathways were negatively enriched upon TCEAL1 knockdown, consistent with its function as a transcription elongation factor in modulating RNA polymerase II–mediated transcription of target genes (30). TCEAL1 can also repress promotor function, and RNA sequencing revealed that loss of TCEAL1 led to up-regulation of genes that had a profound effect on processes involved in the cell cycle, including target genes of E2F transcription factors and G2M checkpoint genes. Interestingly, one of the most up-regulated E2F target genes is *DSCC1* (DNA Replication and Sister Chromatid Cohesion 1). Deletion of a *DSCC1* yeast homologue (*Dcc1p*) resulted in severe sister chromatid cohesion defects, and importantly, increased sensitivity to microtubule depolymerising drugs (31). In addition, overexpression of the *ESPL1* separase protease (another E2F target gene) was observed upon TCEAL1 knockdown; ESPL1 is implicated to increase aneuploidy in a murine breast cancer model (32). Furthermore, in the GSEA, mitotic spindle genes were positively enriched for all treatments (Fig 4B), revealing that TCEAL1 loss, as well as docetaxel treatment, is altering mitotic microtubule dynamics, which may also be important in affecting mitotic progression. Combined, this evidence points towards a role for TCEAL1 in the cell cycle.

Flow cytometry of docetaxel treated cells showed an increase in S phase, G2M and polyploidy consistent with stabilisation of microtubules by taxanes. TCEAL1 siRNA alone had a lesser effect; however, there was a small but significant decrease in G1 and increase in polyploidy, in line with our transcriptomic data on TCEAL1-mediated effects on the cell cycle. Combined *TCEAL1* siRNA and docetaxel appeared to have an effect that was distinct from the individual treatments and control cells, where sub-G1 cells and polyploidy were potently increased. Specifically, we identified E2F targets and genes involved in G2M regulation to be involved after combined TCEAL1 silencing and docetaxel treatment.

Collectively, our whole genome in vivo CRISPR screen has identified TCEAL1 as a potential target to sensitise prostate cancer cells to docetaxel. Future in vivo studies would focus on tumour response to treatment and further work would be warranted to decipher the mechanism by which TCEAL1 regulates the cell cycle, thus allowing the development of a more precise approach for combination treatment with docetaxel. In addition, because docetaxel is often combined with ADT as an upfront treatment for routine

---

in whole cell lysates from SP1 cells transfected with Cas9-EGFP. α-tubulin is used as a loading control. **(G)** Western blot images to confirm expression of Cas9 cytoplasmic and nuclear extracts from FACS-sorted SP1 cells with stable Cas9-EGFP expression. Lamin B and α-tubulin were used as nuclear and cytosolic markers, respectively. **(H)** Schematic illustration of in vivo CRISPR/Cas9 screen. SP1 cells were stably transfected with Cas9-EGFP. After double FACS sorting, SP1 cells with stable expression of Cas9 were selected and amplified for the screen. GeCKO2 V2 whole genome sgRNA library A was used for lentiviral production and transduction of SP1 Cas9-EGFP cells. After 7 d of puromycin selection, the infected SP1 cells were injected in the anterior prostates of CD1-immunocompromised mice. After 7 d of recovery, mice were randomised and treated with vehicle (n = 9) or docetaxel (n = 5). **(I)** sgRNA transfected SP1 prostate orthograft burden in CD-1 nude immunocompromised mice treated as indicated (Vehicle, n = 9; docetaxel, n = 5; ns, not significant; Mann–Whitney test; mean values ± SD are shown). Source data are available for this figure.

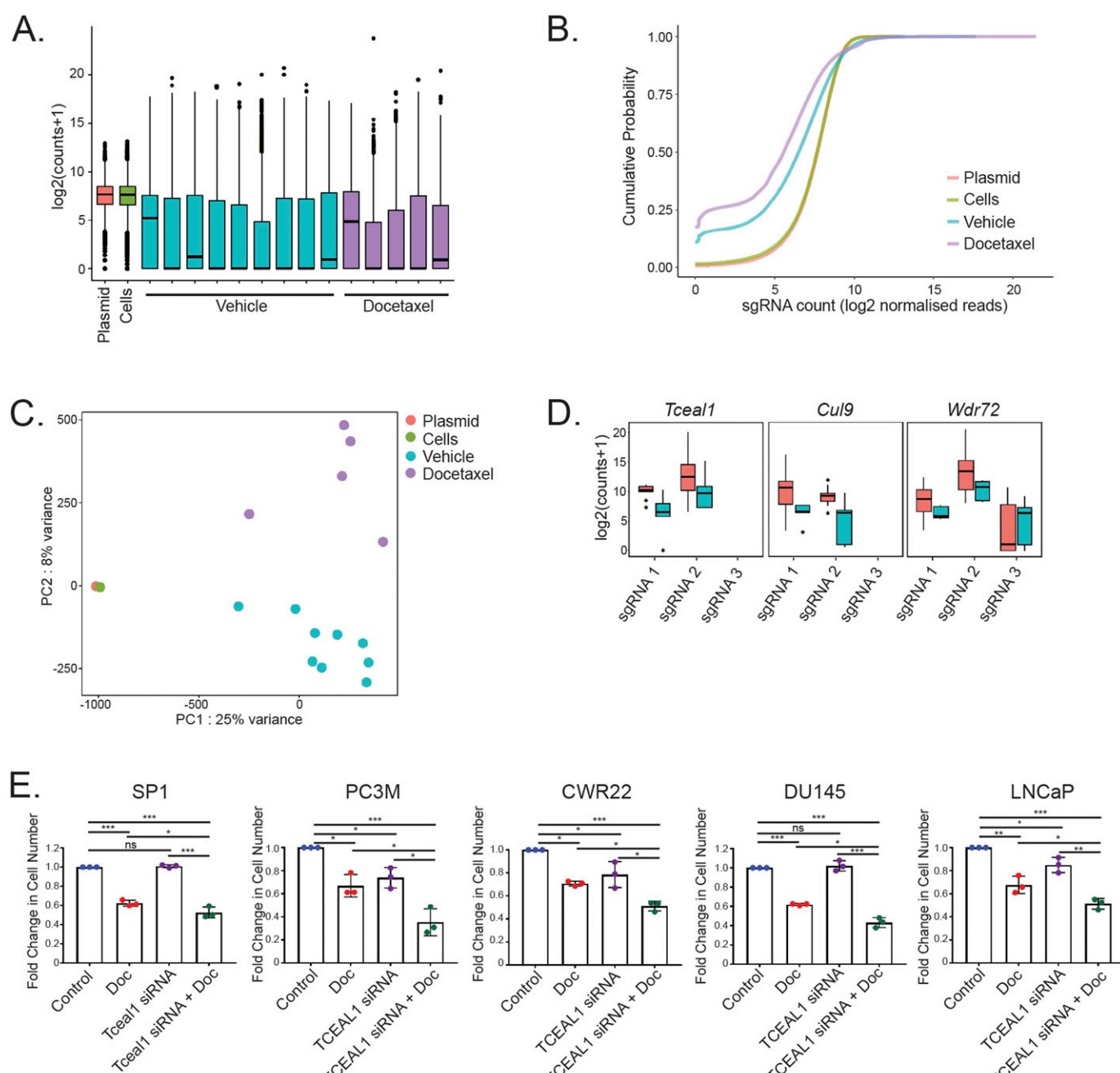

**Figure 2. Bioinformatics analysis identifies negatively selected genes.**
**(A)** Boxplot of the sgRNA-normalised read counts for the plasmid, pre-injection cells, and vehicle and docetaxel-treated tumour samples. Summary statistics shown are median, hinges for the 25th and 75th percentiles, whiskers extending from the hinges to the smallest/largest value no further than 1.5 × IQR from the hinge and "outlying" points. **(B)** Cumulative probability distribution of sgRNAs in the plasmid, pre-injection cells, and vehicle and docetaxel-treated tumour samples. Shift in the curves for vehicle and docetaxel-treated tumour samples represents the depletion in a subset of sgRNAs after injection and after injection and docetaxel treatment, respectively. Distributions for each condition are averaged across replicates. **(C)** Principle component analysis plot of plasmid (n = 1), cells (n = 1), and vehicle (n = 9) and docetaxel (n = 5)-treated tumour samples. Each dot represents one primary prostate tumour from the respective experimental groups. **(D)** Boxplot of sgRNA normalised read counts for each sgRNA detected for three selected significant (padj < 0.25) negatively selected genes in the mock and docetaxel treated samples. Summary statistics shown are median, hinges for the 25th and 75th percentiles, whiskers extending from the hinges to the smallest/largest value no further than 1.5 × IQR from the hinge and "outlying" points. **(E)** The indicated cell lines were transfected with non-targeting or targeting siRNA for 24 h before treatment with DMSO or docetaxel for a further 72 h. The number of cells was counted and the fold change compared with control is shown (n = 3 independent biological experiments, with three independent wells; *P < 0.05, **P < 0.001, ***P < 0.0001; one-way ANOVA with Tukey's test; mean values ± SD are shown).

**Table 1. Significant (padj < 0.25) negatively selected genes.**

| Gene symbol | Detected sgRNAs | Good sgRNAs | log$_2$ fold change | Adjusted *P*-value |
|---|---|---|---|---|
| *Tceal1* | 2 | 2 | −3.4 | 0.0267 |
| *Gm10921* | 3 | 2 | −2.6 | 0.0324 |
| *Gm10058* | 3 | 3 | −1.2 | 0.0324 |
| *mmu-mir-466o* | 2 | 2 | −2.4 | 0.0324 |
| *Vmn1r100* | 2 | 2 | −2.5 | 0.0324 |
| *Cul9* | 2 | 2 | −4.3 | 0.0364 |
| *0610010B08Rik* | 3 | 3 | −1.9 | 0.0526 |
| *Defa25* | 2 | 2 | −1.5 | 0.0525 |
| *Gm14288* | 2 | 2 | −4.4 | 0.0525 |
| *mmu-mir-669d* | 2 | 2 | −3.5 | 0.0583 |
| *Olfr522* | 2 | 2 | −2.4 | 0.0642 |
| *Mettl10* | 2 | 2 | −4.9 | 0.0675 |
| *5031410I06Rik* | 3 | 3 | −3.6 | 0.1235 |
| *Ccl21a* | 3 | 3 | −1.3 | 0.1848 |
| *Wdr72* | 3 | 2 | −2.2 | 0.1848 |
| *Hist1h2bc* | 2 | 2 | −1.8 | 0.1848 |
| *Gm2913* | 3 | 2 | −4.4 | 0.2331 |

Genes in bold have identifiable human orthologues. The number of detected sgRNAs (library A contained three sgRNAs for each gene and four sgRNAs per miRNA) is shown. "Good" sgRNAs is the number of detected sgRNAs that were negatively selected.

management of metastatic prostate cancer, future studies to test the value of TCEAL1 in the context of combined chemo-hormonal therapy are necessary. Whereas the therapeutic landscape of systemic treatment for advanced prostate cancer has changed significantly with the successful introduction of AR pathway inhibitors, taxane chemotherapy remains to have a key role in the management of patients with incurable disease. Cabazitaxel is a second-line taxane chemotherapy which is administered when resistance to docetaxel emerges. Although cabazitaxel and docetaxel use the same mechanism of action in stabilising polymerised microtubules leading to cell death, cabazitaxel is able to by-pass the multidrug resistance (MDR) proteins. Exploring mRNA expression data after TCEAL1 knockdown, changes in the expression of MDR genes are unlikely to be responsible for the enhanced effects of docetaxel treatment (Table S4). In future studies, it would therefore be pertinent to test if suppression of TCEAL1 expression also sensitises prostate cancer cells to cabazitaxel treatment. Besides prostate cancer, docetaxel is also used in a range of cancer types, including breast, stomach, head and neck, and non-small cell lung cancer. Sensitising cancer cells to docetaxel by targeting TCEAL1-mediated mechanism could therefore have wider implications for cancer therapy.

## Materials and Methods

### Cell culture

SP1 cells were derived from a genetically engineered mouse prostate cancer model (SP: *Probasin-Cre Pten$^{fl/fl}$ Spry2$^{fl/+}$*) that represents the loss of Pten tumour suppressor protein and inactivation of Sprouty2 as described in references 12 and 13 (RRID: CVCL_VQ86). Cells were grown in DMEM supplemented with 10% FBS and 2 mM L-glutamine. PC3M, LNCaP, DU145, CWR22, and RWPE human prostate cancer cells were obtained from American Type Culture Collection. PC3M, LNCaP, and DU145 cells were grown in RPMI-1640 supplemented with 10% FBS and 2 mM L-glutamine. RWPE cells were grown in keratinocyte medium supplemented with EGF and bovine pituitary extract. CWR22 cells were grown in RPMI-1640 without phenol red supplemented with 10% charcoal stripped serum and 2 mM L-glutamine. All cell lines used were tested 6 mo for mycoplasma using an in-house MycoAlert Mycoplasma Detection Kit (Lonza), according to the manufacturer's instructions.

### Establishment of docetaxel treatment schedule

The in vivo experiments were carried out in accordance with the UK Home Office regulations (UK Animals [Scientific Procedures] Act 1986) under Project Licence P5EE22AEE.

SP1 cells were orthotopically injected into the anterior prostate of nine 6-wk-old male CD-1 nude mice. Mice were monitored by ultrasound 10 d after surgery to detect tumour formation, before randomisation and the start of treatment (vehicle n = 5, docetaxel n = 4). Mice were treated with either 6 mg/kg docetaxel or vehicle control by intraperitoneal injection every 4 d. The clinical end points for this study were tumour diameter greater than 1.2 cm, tumour invasion into other organs including the bladder, and abdominal distension. Mice were monitored by ultrasound imaging and were euthanized when they reached the clinical end point. After euthanasia of the animals, the prostate orthografts were harvested for immunohistochemical analysis.

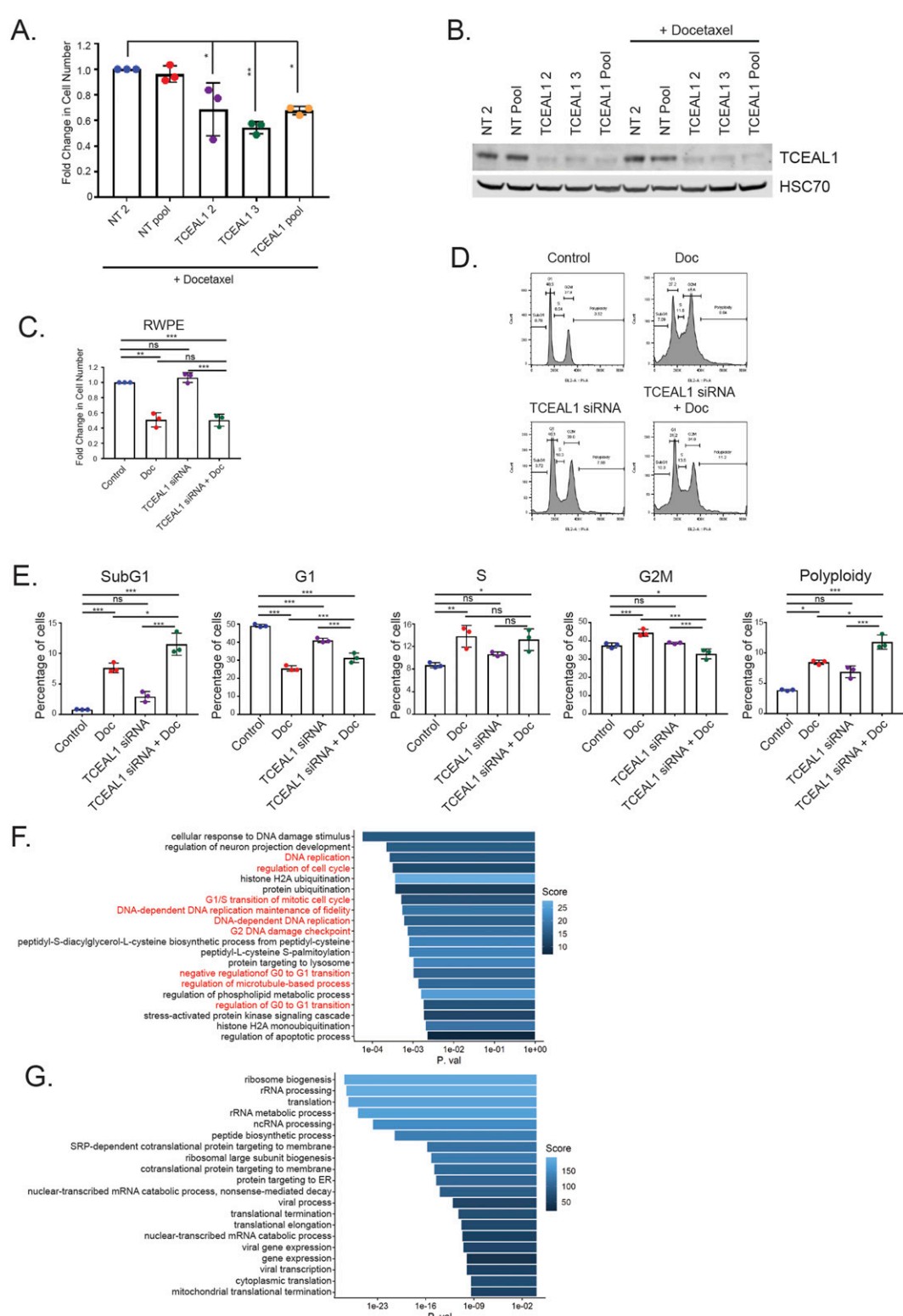

**Figure 3. Analysis of TCEAL1 knockdown–mediated effects.**
**(A)** PC3M cells were transfected with non-targeting (NT2, NT pool) or *TCEAL1*-targeting (individual [TCEAL1 2 and TCEAL1 3] or pooled [TCEAL1 pool]) siRNA as indicated for 24 h before treatment with docetaxel for a further 72 h. The number of cells was counted and the fold change compared with NT2 control is shown (n = 3 independent biological experiments, with three independent wells; *$P < 0.05$, **$P < 0.001$; one-way ANOVA with Dunnett's test; mean values ± SD are shown). **(A, B)** PC3M cells were treated as panel (A). Western blot image of TCEAL1 expression after siRNA transfection. HSC70 is used as a loading control (n = 3, a representative blot is shown). **(C)** RWPE cells were transfected with non-targeting or *TCEAL1*-targeting pooled siRNA as indicated for 24 h before treatment with DMSO or docetaxel for a further 72 h. The number of

## Immunohistochemistry

Immunohistochemistry (IHC) for Ki67 was performed on formalin-fixed paraffin-embedded sections from mouse prostate tumours using Dako Autostainer as previously described (33). Ki67 antibody (RM-9106; Thermo Fisher Scientific) was used at 1:100, with Dako Envision anti-rabbit secondary reagent (K4003; Agilent).

## CRISPR library

We used the two-vector murine CRISPR knockout GeCKOv2 pooled library from Addgene (11). The complete library contains 130,209 different sgRNA sequences targeting 20,611 different genes, as well as 1,175 miRNAs, and is divided into libraries A and B. The sgRNAs in library A (containing three different gRNA sequences per gene and four different sgRNA sequences per miRNA, as well as 1,000 non-targeting controls), designed to have minimal homology to sequences in the mouse genome, were used in the screen.

## Generation of Cas9-expressing cell line

For the genome-wide CRISPR screen, Cas9 expressing SP1 cells were generated by transfecting the cells with a Cas9-EGFP (lenti-Cas9-NLS-FLAG-2A-EGFP; 63592; Addgene) (11) plasmid using nucleofection. The plasmid contains a P2A sequence between the Cas9 and GFP, and so GFP serves as a surrogate marker for Cas9 expression. We checked for expression of Cas9 by Western blot, before the Cas9-EGFP–expressing SP1 (SP1-Cas9) cells were enriched by double consecutive live cell sorting for EGFP-positive cells using the BD FACSAria (BD Biosciences). We further checked the sorted SP1-Cas9 cells for nuclear Cas9 expression before viral transduction.

## Lentivirus production and cell transduction

The GeCKOv2 library was amplified and used to produce lentivirus. After production, the lentivirus was titered and the Cas9 expressing–SP1 cells were transduced with a multiplicity of infection less than 0.4. SP1-Cas9 CRISPR cells were maintained under puromycin selection (9 μg/ml) to select for cells expressing a gRNA and provide time for gene editing to occur. After 9 d under selection, the cells were collected and injected into the mice as described below (3 × 10⁷ cells were removed before the start of injections and were frozen for genomic extraction and used as a reference baseline to indicate which sgRNA sequences were present before the start of the screen).

## In vivo CRISPR screen

10⁷ SP1-Cas9 CRISPR cells were orthotopically injected into the anterior prostate of each 6-wk-old male CD-1 nude mouse. Mice were monitored for tumour burden by ultrasound imaging and treatment started 7 d after surgery. The mice were randomised and treated with vehicle (n = 9) or docetaxel (6 mg/kg, n = 5). Ultrasound imaging allowed monitoring of tumour growth. Overall, all experimental mice received three injections of 6 mg/kg docetaxel administered by intraperitoneal injection before the first mice reached pre-determined clinical end point such as abdominal distension. At this point, all prostate tumours were harvested and finely ground. DNA was extracted from each whole tumour using the Blood and Cell Culture DNA Maxi kit (QIAGEN) according to the manufacturer's instructions.

## DNA preparation and deep sequencing

DNA was prepared for deep sequencing by conducting a two-step PCR. The initial PCR amplified a region of the gRNA cassette to maintain library representation, whereas the second PCR added the primers required for sequencing. DNA was extracted from 100 mg of ground tumour (or entire tumour if weight under 100 mg) using QIAamp DNA Mini Kit (QIAGEN) as per the manufacturer's instructions. Each column takes a maximum of 25 mg of tissue; tumours were split into 5 × ~20 mg tissue, and extracted DNA was combined at the end. PCR was repeated 35 times for 100-fold library representation (primer sequences are the same as those used in Chen et al [2015] (11)). All PCR products per sample were combined and used in the second PCR round to add the primers required for sequencing. PCR products from each sample were again combined and concentrated using a QIAquick PCR Purification Kit (QIAGEN) as per the manufacturer's instructions. Each sample was run on a 1.5% agarose gel. Bands were excised and DNA purified using a QIAquick Gel Extraction Kit before sending for sequencing. The samples were deep sequenced using the Illumina platform. The resulting data were de-multiplexed and analysed by bioinformatics to identify genes important in the response to docetaxel.

## Bioinformatic analysis for CRISPR screen

Sequencing reads were first trimmed using cutadapt (v2.5) (34) to obtain the 20-bp spacer (guide) sequences. The initial quality control of sequencing data before and after trimming was performed using FastQC (v0.11.4) (35). The spacer sequences were then mapped, quantified, and analysed using various functions from the

cells was counted and the fold change compared with control is shown (n = 3 independent biological experiments, with three independent wells; ***P < 0.0001, **P < 0.001, ns, not significant; one-way ANOVA with Tukey's test; mean values ± SD are shown). **(D)** Cell cycle profiles of PC3M cells treated as indicated. Cells were transfected with either control (non-targeting) or *TCEAL1*-targeting pooled siRNA for 24 h before being synchronised by a double thymidine block. Cells were released into fresh media containing DMSO or docetaxel for 48 h. All cells were collected and fixed in ethanol. After fixation, cells were stained with propidium iodide and analysed using flow cytometry (n = 3 independent biological experiments, representative plots are shown). **(E)** Quantification of percentage of PC3M cells in all stages of the cell cycle as indicated (n = 3 independent biological experiments; *P < 0.01, **P < 0.001, ***P < 0.0001; one-way ANOVA with Tukey's test; mean values ± SD are shown). **(F)** Plot showing the top 20 enriched Gene Ontology biological processes for genes up-regulated upon TCEAL1 suppression. The colour of the bar details the enrichment score, and the x-axis is the *P*-value. Processes involved in the cell cycle are highlighted in red. **(G)** Plot showing the top 20 enriched Gene Ontology biological processes for genes down-regulated on TCEAL1 suppression. The colour of the bar details the enrichment score, and the x-axis is the *P*-value.
Source data are available for this figure.

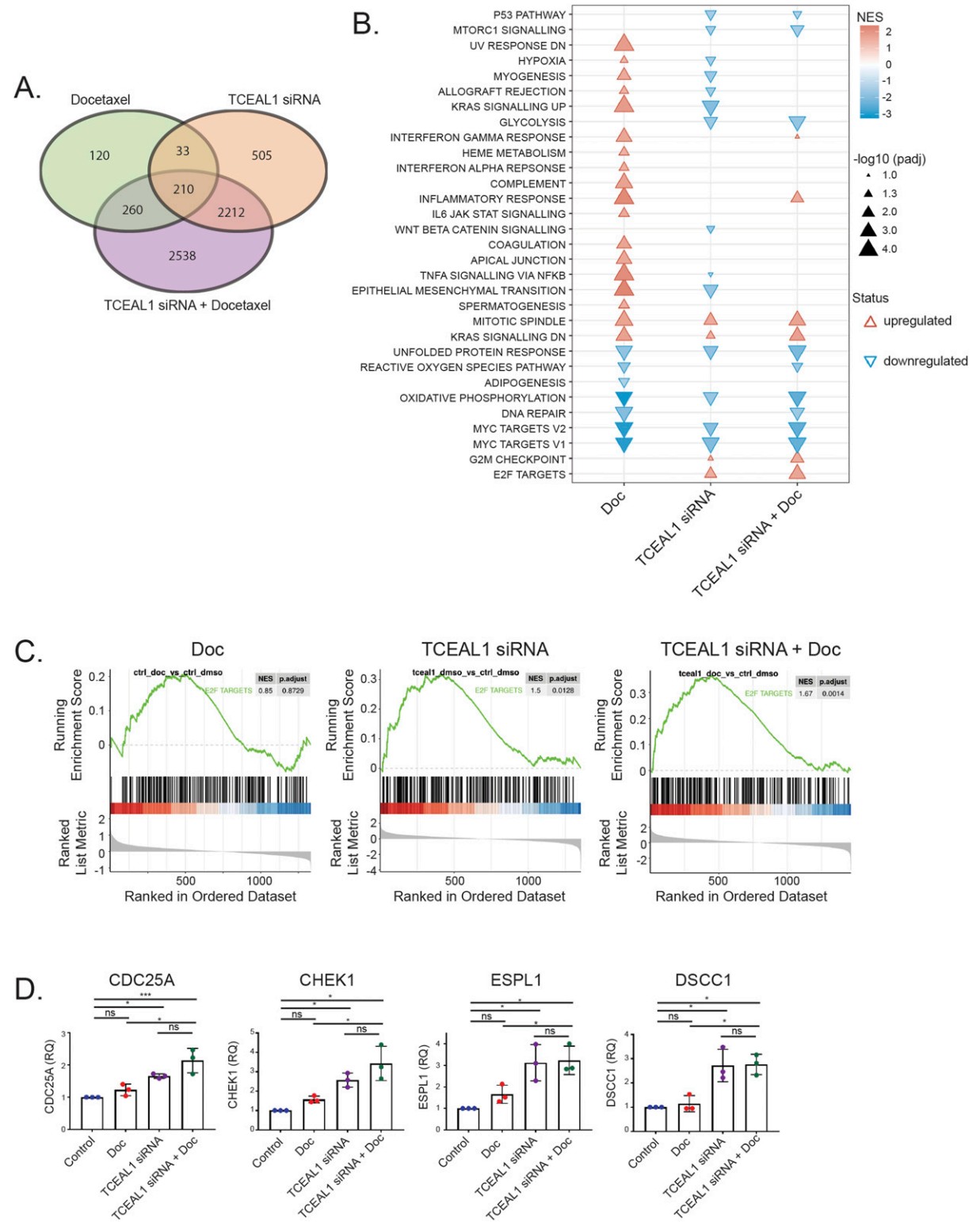

**Figure 4.  Transcriptome informed pathway analysis upon suppressed TCEAL1 expression combined with docetaxel treatment.**
**(A)** PC3M cells were transfected with non-targeting or *TCEAL1*-targeting pooled siRNA for 24 h before treatment with DMSO or docetaxel for a further 48 h. RNA was extracted and sequenced. Venn diagram shows the number of genes that had altered expression in the three treatment conditions compared with control samples (n = 4 independent biological experiments). **(B)** Plot showing the enriched gene sets after Gene Set Enrichment Analysis from RNA sequencing using the Hallmark gene sets. X-axis shows the sample condition, with the enriched gene sets on the left of the plot. The legend details triangle size relative to −log$_{10}$ of the adjusted *P*-value (1.3 = −log$_{10}$0.05). Colour shows the Normalised Enrichment Score (NES) compared with the control (non-targeting siRNA and DMSO) (Doc = docetaxel treatment). **(C)** Enrichment

Model-based Analysis of Genome-wide CRISPR/Cas9 (MAGeCK) (v0.5.6) (36) tool and using the robust ranking aggregation algorithm. Collected sgRNA read counts were normalised by total read counts (–norm-method total) and only sgRNAs with an average expression higher than 100 reads across the treatment groups (either vehicle or docetaxel), and genes with at least two sgRNAs detected were kept for further analysis. A depletion/enrichment analysis was performed using MAGeCK test command with additional parameters (-norm-method total; –adjust-method fdr; –additional-rra-parameters "--min-percentage-goodsgrna 0.6"), to re-normalise raw counts after filtering, and filter genes that have a low percentage of "good sgrnas" (sgRNAs whose ranking is below the $\alpha$ cut-off). Data analysis was performed in R (v3.6.1) (37) using packages dplyr (v0.8.3) (38), tidyr (v1.0.0) (39), and tibble (v2.1.3) (40). Figures were generated using ggplot2 (v3.2.1) (41), pheatmap (v1.0.12) (42), ggpubr (v0.2.3) (43), and kableExtra (v1.1.0) (44). The code for pre-processing and data analysis is available to view at https://github.com/prepiscak/optichem_crispr.

Potential off-target effects for each of the three *Tceal1* sgRNAs used were examined using https://wge.stemcell.sanger.ac.uk/find_off_targets_by_seq with Mouse (GRCm38) and PAM Right (NGG) (Table S4).

### siRNA transfection

Cells were transfected with either non-targeting or targeting siRNA (25 nM) using Lipofectamine RNAiMax (Life Technologies) according to the manufacturer's instructions before subsequent treatment and analysis. The following siRNAs from Dharmacon were used: ON-TARGETplus Mouse *Tceal1* SMARTPool; ON-TARGETplus Human *TCEAL1* SMARTPool; ON-TARGETplus Human *Tceal1* Set of four siRNAs; ON-TARGETplus Mouse *Cul9* SMARTPool; ON-TARGETplus Human *CUL9* SMARTPool; ON-TARGETplus Mouse *Wdr72* SMART-Pool; ON-TARGETplus Human *WDR72* SMARTPool; ON-TARGETplus non-targeting pool; ON-TARGETplus non-targeting control siRNA 1; ON-TARGETplus non-targeting control siRNA 2.

### RNA extraction

Total mRNA was extracted using the RNeasy Mini Kit (QIAGEN) according to the manufacturer's instructions. RNA was quantified using the NanoDrop 2000 spectrophotometer (Thermo Fisher Scientific). For RNA sequencing samples, RNA quality was assessed on a 2100 Bioanalyser (Agilent).

### Quantitative real-time PCR

cDNA was prepared using the High Capacity cDNA Transcription Kit (Applied Biosystems) according to the manufacturer's instructions, and Taqman qRT-PCR was performed as previously described (33).

### Cell cycle analysis

Cells were seeded and reverse transfected with siRNA 24 h using Lipofectamine RNAiMax (Life Technologies) before synchronising using a double thymidine (2 mM) block. Cells were released into fresh medium before treatment with docetaxel (2 nM) or DMSO. After 48 h, all cells (floating and attached) were harvested and fixed in 70% ethanol for at least 1 h. Cells were washed with PBS before incubation with RNase A and propidium iodide for 30 min. Samples were analysed on an Attune NxT Flow Cytometer (Thermo Fisher Scientific). Data were analysed using FlowJo software, and the percentage of cells in each phase of the cell cycle was determined.

### Western blot

Whole-cell lysates from PC3M cells were prepared by lysing cells in lysis buffer (20 mM Hepes, 0.5 mM EGTA, 0.5% NP40, and 150 mM NaCl with protease and phosphatase inhibitors). Lysates were resolved by SDS–PAGE on 4–12% gradient Bis-Tris gels (Life Technologies) before wet transfer to PVDF membrane (Millipore) using the NuPage transfer module (Life Technologies). Membranes were blocked with 5% milk before incubation with primary antibody overnight at 4°C. After incubation with secondary antibodies, Alexa Fluor 680 goat anti-rabbit (Life Technologies) or goat anti-mouse DyLight 800 (Thermo Fisher Scientific) bands were visualised using the LI-COR (LI-COR Biosciences). Primary antibodies used were TCEAL1 (sc-393621, 1:200; Santa Cruz Biotechnology) and HSC70 (sc-7298, 1:1,000; Santa Cruz Biotechnology).

### RNA sequencing and bioinformatics

RNA from PC3M cells was isolated and quantified as described above. Libraries from these samples were prepared for sequencing using the Illumina TotalPrep RNA Amplification Kit (Ambion, Life Technologies) with Poly(A) selection according to the manufacturer's instructions. Quality checks and trimming on the raw RNA-Seq data files were conducted using FastQC version 0.11.7 (35), FastP (45), and FastQ Screen version 0.12.0 (46). RNA-Seq paired-end reads were aligned to the Ensembl version 38 build 95 (47) of the human genome and annotated using HiSat2 version 2.1.0 (48). Expression levels were determined and were statistically analysed by a combination of the following: HTSeq version 0.9.1 (49); the R environment version 3.5.3 (37); packages from the Bioconductor data analysis suite (50); and differential gene expression analysis based on the negative binomial distribution using the DESeq2 package version 1.22.2 (51). Functional enrichment analysis was conducted with enrichR R package (v2.1) (52, 53) to the GO BP 2018 database. GSEA on Hallmarks gene set collection (54) was carried out using clusterProfiler (v3.12.0) (55) and fgsea (v1.10.1) (56 *Preprint*) algorithm with genes from RNA-seq differential expression analysis ranked according to the $\log_2$ fold change and converted to

---

plots of Hallmark E2F target genes (gene set size = 200) for each of the indicated treatment conditions (n = 4 independent biological experiments; NES, Normalised Enrichment Score, *P*.adjust = a Benjamini–Hochberg adjusted *P*-value). **(D)** qRT-PCR validation of selected E2F target genes. PC3M cells were transfected with non-targeting or *TCEAL1*-targeting pooled siRNA as indicated for 24 h before treatment with DMSO or docetaxel for a further 72 h. *Casc3* was used as a reference gene for normalisation, and the fold change compared with control is shown (RQ, relative quantitation; n = 3 independent biological experiments; *P < 0.05, ***P < 0.0001; one-way ANOVA with Tukey's test; mean values ± SD are shown).

entrez_gene_ids using Ensembl Genes 96 annotation. Data analysis was performed in R (v3.6.1) (37) and figures were generated using combination of ggplot2 (v3.2.1) (41) and enrichplot (1.4.0) (57).

## Statistical analysis

Data plotting and statistical analyses including one-way ANOVA with Tukey's test, Welch's *t* test (unpaired, two tailed), Mann–Whitney, Kaplan–Meier survival analysis, and log-rank (Mantel–Cox) were carried out using GraphPad Prism 7. Graphs are shown as mean ± SD with individual points shown. *P*-values for all experiments and statistical tests are shown in Table S5.

# Data Availability

CRISPR screen and RNA-seq data have been deposited in the ArrayExpress database at EMBL-EBI (www.ebi.ac.uk/arrayexpress) under accession numbers E-MTAB-9482 and E-MTAB-9484, respectively.

# Supplementary Information

# Acknowledgements

We thank Arnaud Blomme and George Skalka for helpful discussions. This work was supported by the Prostate Cancer Foundation Challenge Award, Cancer Research UK (A17196: core funding to the Cancer Research UK Beatson Institute, A15151 and A22904: awarded to HY Leung, and A29252: awarded to M Bushell).

## Author Contributions

LK Rushworth: conceptualization, data curation, formal analysis, validation, investigation, visualization, methodology, and writing—original draft, review, and editing.
V Harle: conceptualization, data curation, formal analysis, validation, investigation, and methodology.
P Repiscak: conceptualization, data curation, software, formal analysis, investigation, and methodology.
W Clark: data curation, formal analysis, and methodology.
R Shaw: data curation, software, formal analysis, and investigation.
H Hall: software, formal analysis, and investigation.
M Bushell: resources and formal analysis.
HY Leung: conceptualization, resources, data curation, formal analysis, supervision, funding acquisition, investigation, methodology, project administration, and writing—original draft, review, and editing.
R Patel: conceptualization, data curation, formal analysis, supervision, funding acquisition, investigation, methodology, and project administration.

## Conflict of Interest Statement

The authors declare that they have no conflict of interest.

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
