## [Reviewer comments · Life Science Alliance]

Life Science Alliance

In vivo CRISPR/Cas9 knockout screen: TCEAL1 silencing enhances docetaxel efficacy in prostate cancer

Linda Rushworth, Victoria Harle, Peter Repiscak, William Clark, Robin Shaw, Holly Hall, Martin Bushell, Hing Leung, and Rachana Patel

DOI: <https://doi.org/10.26508/lsa.202000770>

Corresponding author(s): Hing Leung, Beatson Institute for Cancer Research

Review Timeline:

Submission Date:	2020-05-08
Editorial Decision:	2020-06-15
Revision Received:	2020-08-28
Editorial Decision:	2020-09-17
Revision Received:	2020-09-22
Accepted:	2020-09-25

Scientific Editor: Shachi Bhatt

Transaction Report:

June 15, 2020

Re: Life Science Alliance manuscript #LSA-2020-00770-T

Prof. Hing Y Leung
Beatson Institute for Cancer Research
Urology Group
Urology
Glasgow, Outside the United States or Canada G611BD
UNITED KINGDOM

Dear Dr. Leung,

Thank you for submitting your manuscript entitled "In vivo CRISPR/Cas9 screen identifies TCEAL1 as a target to enhance docetaxel efficacy in prostate cancer" to Life Science Alliance. The manuscript was assessed by expert reviewers, whose comments are appended to this letter.

The study is generally well received, yet the reviewers point out some technical issues and questions that should be addressed. For the in vivo validation using xenograft models (point 1 of reviewer #3), while of course such data would be very welcome, we understand that this may be out of scope for this study and would not make it a strict requirement.

Thank you for this interesting contribution to Life Science Alliance. We are looking forward to receiving your revised manuscript.

Sincerely,

Reilly Lorenz
Editorial Office Life Science Alliance
Meyerhofstr. 1
69117 Heidelberg, Germany
t +49 6221 8891 414
e contact@life-science-alliance.org
www.life-science-alliance.org

B. MANUSCRIPT ORGANIZATION AND FORMATTING:

Reviewer #1 (Comments to the Authors (Required)):

Well described and executed study;
one major limitation should be addressed, i.e. the histology and immunophenotyping of the mouse model;
1. adenocarcinoma? 2. AR expression? 3. CK18 expression?
if the (immuno)phenotype is not similar to human PrCa, that should be addressed as a limitation

Reviewer #2 (Comments to the Authors (Required)):

The paper by Rushworth et al., provides an exciting advance in our understanding of factors influencing response of prostate cancer cells to docetaxel treatment which will be of future significance in the clinical setting. The study undertook the first in vivo CRISPR screen in prostate cancer using a relevant murine model developed by the same group to identify gene knockouts that sensitised cells to docetaxel-mediated cell kill in an effort to provide new targets for combination therapies involving taxanes.

This paper is of interest to the community and has used some exciting approaches to identify TCEAL1 as a regulator of Doc sensitivity in prostate cancer cells. There is a combination of in vivo and cell line-based experimentation which are pretty complimentary in their demonstration that TCEAL1 depletion is a potentiating factor in Doc treatment sensitivity, although there are some indications in the data where this does not really show through and is discussed below. The discussion section is nicely composed and indicates what type of studies are needed to fully understand the role of TCEAL1 interplay with Doc treatment. I would say that this is certainly required as I felt that a weakness of the paper was the lack of mechanistic insight into what TCEAL1 was actually doing in cells to govern response to doc. Another interesting set of experiments would be to apply cabazitaxel in the same cell line experiments to understand if the effect of TCEAL1 is consistent across the different taxane-based chemotherapeutics.

My other comments are listed below:

1. Can the authors state whether in Figure 1, was the Cas9 blot from cells transfected just with Cas9 or with Cas9 + gRNA? It is interesting to note the considerable cytoplasmic presence. How many NLS sequences does this protein have? Also, in keeping with the Cas9 comments, can the authors state specifically which Cas9 vector was used as I am not able to distinguish whether the Cas9 is directly fused to GFP, or whether there is a T2A sequence between them so that GFP is simply a surrogate marker for Cas9 expression.
2. For the description of the methods section, it is a little unclear how long after the cells were transduced before being implanted into the host. Also, were they put under selection beforehand and for how long.
3. Formatting for Figure 1H needs adjusting as it is difficult to make out the 10^7 .
4. The description of the potential false negative issues surrounding the library versus pre-implant cell sgRNA coverage is informative. Have the authors selected any essential genes to study by QRT-PCR to see if they have been diminished by the sgRNAs?
5. Is there a pipeline for identifying potential off-target effects of the individual gRNAs for the chosen targets (e.g TCEAL1 etc) to rule out any possibility that the effects seen in the screen is not by KO of such an off-target sequence?
6. Is there any evidence to suggest that TCEAL1 depletion in Doc-resistant models would have a re-sensitising effect? Is there any clinical data to suggest that in Doc-resistant tumours that TCEAL1 levels are altered?

7. In Figure 3A, it is difficult to unpick the effect of TCEAL1 KD on Doc sensitivity because the effect of KD seems to be the same in both the presence and absence of Doc.

8. What is the expression of TCEAL1 in RWPE cells? In the FACS data, can the authors investigate if there is a synergistic or additive effect of KD and Doc treatments as it seems from eye-balling the data that the SubG1 level for the dual treatment/KD is the sum of KD and Doc alone.

Reviewer #3 (Comments to the Authors (Required)):

Summary

The manuscript by Rushworth et al. centers around the results of an in vivo CRISPR knockout screen to identify candidate genes that act to limit the effects of (clinically relevant) docetaxel treatment on prostate cancer. The authors derived a murine cell line (SP1) from prostatic tumors in probasin-Cre Pten^{fl/fl} Spry2^{fl/+} mice that were subsequently modified to express CAS9 to enable CRISPR-based targeting of genes using a library of targeted guide RNAs. The modified SP1 cells were then implanted in the prostates of CD1 nude mice to form tumors and randomized into control and docetaxel treatment groups. Depleted guide RNAs were identified by bioinformatic analysis of sequencing reads of PCR-amplified regions of genomic DNA. The screen identified several candidate genes that the authors further examined by reducing their expression in cultured human prostate cancer cells using siRNA and assessing the resultant enhancement of the effects of docetaxel. The authors primarily focus on the top candidate TCEAL1 and further examined gene expression changes induced by decreasing its expression with siRNA in the absence and presence of docetaxel. The validation studies are relatively well described and provide reasonable evidence to support the authors' claims. The manuscript is well written, albeit generally descriptive in nature, providing sufficient evidence to implicate TCEAL1 and other potential candidate genes as modulators of docetaxel effects in prostate cancer.

Major critiques

1) In vivo versus in vitro effects. The authors make a major case for the importance of an in vivo screen to identify modulators of docetaxel effects on prostate cancer; however, all validation of the screen hits was performed in prostate cancer cell lines in culture. The authors do not provide any evidence that the results of the screen could be recapitulated in vivo. The importance of TCEAL1 as a modulator of docetaxel would be significantly strengthened by introducing the same sgRNAs targeting TCEAL1 (and relevant control sgRNAs) in SP1 cells and assessing the ability of knocking out this gene on docetaxel treatment of xenografted tumors. The authors suggest that reducing TCEAL1 gene expression enhances the effects of docetaxel, which should also equate to prolonged survival (or at least decreased tumor growth) of SP1-tumor-bearing mice. In addition, the authors provided survival data (Kaplan-Meier curves) only for the murine prostate cancer model from which the SP1 cells were derived. The authors state that weekly ultrasounds are used to assess tumor growth but they do not provide any data for the effects of docetaxel on SP1 tumor growth or mouse survival--data that would be more relevant to the genetic screen than the data from the original SP mouse model that is shown in Fig 1.

2) The screen uses CRISPR/Cas9 genetic targeting, which would be expected to result in complete loss of gene expression whereas all validation was performed using siRNA, which dynamically alters expression (i.e. effects are time-dependent) and does not completely eliminate the expression of their target.

3) Neither the data nor the analytical approach (computational tools) were provided or made available online, preventing deeper evaluation of the results. The author-provided link https://github.com/prepiscak/optichem_crispr is not valid (or the repository is not public). The data do not appear to have been deposited into public databases such as EBI or SRA.

Minor critiques

1) The potential role of androgen (dependence of tumor cells and deprivation therapy), a major factor in prostate cancer etiology and prognosis, is only peripherally described. More discussion is warranted.

2) The screen would produce hits for gene targets of the sgRNA library that are both under- and over-represented in the docetaxel-treated tumors compared to control yet only the under-represented genes are described. Similarly, genes essential for tumor growth would also be expected to be depleted in the untreated tumors compared to the pre-injection tumor cells. Since the essential genes are likely not a large proportion of the genes targeted by the library, it is not surprising that a summary statistic of sgRNA sequence abundance does not change between conditions (as depicted in EV2A). Examination of a subset of genes shown to be essential in other published studies could be useful.

3) By the authors' own admission, the statistical power of the screen was limited, possibly due to technical reasons. However, it is possible that the statistical power could be enhanced simply using alternative, more recent analytical tools such as JACKS for the CRISPR knockout analysis (<http://www.genome.org/cgi/doi/10.1101/gr.238923.118>).

4) Although "clinical end point" is used in the manuscript several times, it is never defined.

5) The representation of the sgRNA library is shown as a boxplot distribution in Fig. 2A and independently for each gene in terms of $\log_2(\text{counts}+1)$. Some depiction of how the distribution of unique sgRNA abundances are maintained or altered in the different conditions is warranted. A waterfall plot of the abundances of each sgRNA in the plasmid library (in increasing or decreasing order) could serve as the baseline for comparison.

6) Y-axis labels "RQ" in Fig 4 (and related EV) are not defined.

Reviewer #1 (Comments to the Authors (Required)):

*Well described and executed study;
one major limitation should be addressed, i.e. the histology and immunophenotyping of the mouse model;*

*1. adenocarcinoma? 2. AR expression? 3. CK18 expression?
if the (immuno)phenotype is not similar to human PrCa, that should be addressed as a limitation*

Response: We appreciate Reviewer 1's positive comments, and they raise an interesting point regarding the histology of the tumours from our mouse models.

Previous experiments using our *PbCre Pten^{fl/fl} Spry2^{fl/+}* GEMM have revealed that prostate tumours developed from these mice have an adenocarcinoma phenotype, characterised by glandular differentiation (1). Since adenocarcinoma is the most common type of clinical prostate cancer in humans, this model is clinically relevant. In addition, Western blot (shown below) of the SP1 cells injected to generate *PbCre Pten^{fl/fl} Spry2^{fl/+}* orthografts shows expression of full length AR protein. This blot has now been included as Fig. EV1B in the manuscript.

We have added the above details about the histology of the mouse model in the Results and Discussion section of the manuscript.

Reviewer #2 (Comments to the Authors (Required)):

The paper by Rushworth et al., provides an exciting advance in our understanding of factors influencing response of prostate cancer cells to docetaxel treatment which will be of future significance in the clinical setting. The study undertook the first in vivo CRISPR screen in prostate cancer using a relevant murine model developed by the same group to identify gene knockouts that sensitised cells to docetaxel-mediated cell kill in an effort to provide new targets for combination therapies involving taxanes.

This paper is of interest to the community and has used some exciting approaches to identify TCEAL1 as a regulator of Doc sensitivity in prostate cancer cells. There is a combination of

in vivo and cell line-based experimentation which are pretty complimentary in their demonstration that TCEAL1 depletion is a potentiating factor in Doc treatment sensitivity, although there are some indications in the data where this does not really show through and is discussed below. The discussion section is nicely composed and indicates what type of studies are needed to fully understand the role of TCEAL1 interplay with Doc treatment. I would say that this is certainly required as I felt that a weakness of the paper was the lack of mechanistic insight into what TCEAL1 was actually doing in cells to govern response to doc. Another interesting set of experiments would be to apply cabazitaxel in the same cell line experiments to understand if the effect of TCEAL1 is consistent across the different taxane-based chemotherapeutics.

Response: We thank the Reviewer 2 for the positive and constructive comments. Our manuscript details the first *in vivo* CRISPR screen in prostate cancer under treatment pressure, and validation of the top gene hits confirms the validity of our screen despite some technical challenges that we uncovered during the execution and analysis of data from the screen.

We agree with Reviewer 2's comment about experiments required to explore the underlying molecular mechanism. The next stage of this project will look in closer detail at how the loss of TCEAL1 may enhance docetaxel-mediated effects, and what pathways are controlled by TCEAL1 functions. To do this, stable TCEAL1 CRISPR knockout cell clones will need to be generated. Even in normal circumstances, the generation and relevant validation of cell clones will take many weeks, given the logistic challenge with the current COVID-19 situation, we were unable to initiate efforts to obtain the cell clones and then carry out detailed mechanistic studies. We hope that Reviewer 2 and the Editors will be content with our suggestion that this work is to be considered as part of future investigations.

Cabazitaxel and docetaxel employ the same mechanism of action in stabilising polymerised microtubules, thus disrupting formation of the mitotic spindle and leading to cell death. Unlike docetaxel, cabazitaxel is able to by-pass the multidrug resistance (MDR) proteins, meaning that cells with upregulated MDR expression remains responsive to cabazitaxel while they are resistant to docetaxel. We thank Reviewer 2 for the nice suggestion of testing if suppression of TCEAL1 expression also sensitises prostate cancer cells to cabazitaxel treatment.

We are able to explore data from the RNA sequencing data on cells with suppressed TCEAL1 expression. We wish to test if TCEAL1 knockdown alters the expression of genes encoding for MDR proteins. For instance, if MDR expression is suppressed by TCEAL1 silencing, then one could hypothesise that the enhanced response to docetaxel may be related to reduced efflux of docetaxel, resulting from reduced MDR expression following TCEAL1 knockdown. On the other hand, if TCEAL1 knockdown increased or did not alter MDR expression significantly, then it is likely that silencing of TCEAL1 expression will also sensitise prostate cancer cells to cabazitaxel treatment.

In the Table below, among the detectable *MDR* genes, TCEAL1 knockdown only resulted in small degree of changes in the MDR gene expression, ranging from -1.57 to +1.34. Of note, ABCB7 is the only down regulated gene showing statistical significance following TCEAL1 knockdown. Among the upregulated MDR encoding genes as a result of TCEAL1 silencing, upregulation of ABCB9 and ABCB10 expression (borderline increase at 1.21 and 1.34 fold

respectively) would suggest that the enhanced response was unrelated to the MDR expression following TCEAL1 knockdown.

Given the logistic difficulties with COVID-19, we have incorporated this consideration as future investigations in the Discussion section of the manuscript.

Gene	Mean TCEAL1 siRNA	Mean Control siRNA	Adjusted p value	Fold Change
ABCB1	1	3	#N/A	-1.20
ABCB4	11	8	0.643237006	1.16
ABCB6	104	109	0.843738072	-1.04
ABCB7	1535	2428	6.61924E-10	-1.57
ABCB8	1653	1848	0.083587956	-1.12
ABCB9	508	420	0.008957489	1.21
ABCB10	931	673	0.044105831	1.34
ABCB11	46	46	0.97878136	1.01

My other comments are listed below:

1. Can the authors state whether in Figure 1, was the Cas9 blot from cells transfected just with Cas9 or with Cas9 + gRNA? It is interesting to note the considerable cytoplasmic presence. How many NLS sequences does this protein have? Also, in keeping with the Cas9 comments, can the authors state specifically which Cas9 vector was used as I am not able to distinguish whether the Cas9 is directly fused to GFP, or whether there is a T2A sequence between them so that GFP is simply a surrogate marker for Cas9 expression.

Response: We thank Reviewer 2 for raising this relevant point. The plasmid used for the transfection was a lentiviral vector, lenti-Cas9-NLS-FLAG-2A-EGFP (lentiCas9-EGFP; Addgene 63592) (2). There is a P2A sequence between the Cas9 and GFP, and so GFP serves as a surrogate marker for Cas9 expression. Due to local safety constrictions for cell sorting Containment Level 2 cells, we were unable to use this as a lentiviral plasmid to transduce the SP1 cells and instead used nucleofection to transfect. Details of the Cas9-EGFP plasmid have been added to the methods section.

In Figure 1, the Cas9 blot shows the expression in cells transfected only with Cas9. These blots were done prior to transducing the cells with the library-containing virus to check for the presence of nuclear Cas9. As the reviewer rightly pointed out, there was considerable cytoplasmic Cas9 presence, which we were also interested to see. This plasmid has only one NLS sequence, and perhaps this resulted in relatively high cytoplasmic levels.

2. For the description of the methods section, it is a little unclear how long after the cells were transduced before being implanted into the host. Also, were they put under selection beforehand and for how long.

Response: We thank Reviewer 2 for these pertinent comments. After the cells were transduced, they were put under puromycin (9 µg/ml) selection for 9 days. After this time, 10⁷ cells were injected into the anterior prostate of CD-1 mice. We apologise for any confusion from our description in the methods section. This section has been rewritten and we hope that this is now clear.

3. Formatting for Figure 1H needs adjusting as it is difficult to make out the 10⁷.

Response: We apologise for the oversight with the figure formatting. This has been corrected and the figure is now clearer.

4. The description of the potential false negative issues surrounding the library versus pre-implant cell sgRNA coverage is informative. Have the authors selected any essential genes to study by QRT-PCR to see if they have been diminished by the sgRNAs?

Response: We have not looked at individual essential genes by qRT-PCR, however we did perform gene set enrichment analysis of essential gene sets. We used essential gene sets as defined in CRISPRcleanR package (v0.5) (3) and found that they were not significantly depleted in the pre-injection cells compared to the plasmid (see table below). Contrary to what we expected, among all of the essential gene sets, only the Ribosomal Proteins gene set reached a significant adjusted p value.

We appreciate that not all genes labelled essential within the gene sets will be necessarily be functionally required for SP1 survival as there is likely cell line variability, but given we had a very small percentage of essential genes lost in the pre-injection cells compared to the library, it suggests that the screen performance was less optimal than anticipated.

Gene set	Normalised Enrichment Score	p value	Adjusted p value
RNA_polymerase	-1.27	0.1554	0.3627
Spliceosome	-0.82	0.7351	0.9762
Proteasome	-0.77	0.7825	0.9762
essential	-0.73	0.9762	0.9762
DNA_Replication	0.63	0.9279	0.9762
Ribosomal_Proteins	1.77	0.0068	0.0389

5. Is there a pipeline for identifying potential off-target effects of the individual gRNAs for the

chosen targets (e.g TCEAL1 etc) to rule out any possibility that the effects seen in the screen is not by KO of such an off-target sequence?

Response: Given CRISPR screens usually have multiple (typically 6 or more) gRNAs per gene, off-target effects can be confidently removed during the analysis by selecting hits that are common to different gRNA for the same genes. However, since we used only 3 gRNAs per gene due to restrictions with the cell number we could orthotopically inject, we checked the TCEAL1 gRNAs from the GeCKOv2 library to look for predicted off-target effects.

Potential off-target effects for each of the three Tceal1 sgRNAs used were examined using https://wge.stemcell.sanger.ac.uk/find_off_targets_by_seq with Mouse (GRCm38) and PAM Right (NGG) (4).

Results from the analysis can be access using links in the following table:

sgRNA	sequence	WGE off targets results
MGLibA_53075	CGTATCCGCCCTCAATTCAT	https://wge.stemcell.sanger.ac.uk/crispr/565054205
MGLibA_53076	GTTCGAAGACCGTATTCCCA	https://wge.stemcell.sanger.ac.uk/crispr/565054187
MGLibA_53077	GTCTGAAGATCGTCCTCCGC	https://wge.stemcell.sanger.ac.uk/crispr/565054182

MGLibA_53075 should uniquely target exon2 (ENSMUSE00000389668; X:136709149-136709171) of TCEAL1 and even up to 4 mismatches allowed there are no other exonic regions targeted. At 4 mismatches there are 16 potential non-exonic (intergenic + intronic) targets.

MGLibA_53076 should uniquely target exon2 (ENSMUSE00000389668; X:136709061-136709083) of TCEAL1 and even up to 4 mismatches allowed there are no other exonic regions targeted. At 4 mismatches there are 25 potential non-exonic (intergenic + intronic) targets.

MGLibA_53077 should uniquely target exon2 (ENSMUSE00000389668; X:136709016-136709038) of TCEAL1. Allowing for 3 mismatches there is one potential intergenic target and at 4 mismatches there are 19 non-exonic (Intergenic + Intronic) and 4 exonic targets. At 4 mismatches exonic targets are Wfs1 (wolframin ER transmembrane glycoprotein; 5:36966856-36966878), Fbxo25 (F-box protein 25; 8:13929385-13929407), Cyp51 (cytochrome P450, family 51; 5:4092409-4092431) and Grwd1 (glutamate-rich WD repeat containing 1; 7:45827625-45827647).

In conclusion, two of the three TCEAL1 sgRNAs are uniquely targeting TCEAL1. The third sgRNA could potentially target other genes, though with 3 or 4 mismatches. In our sequencing data however, MGLibA_53077 was only detected in Plasmid, Cells and one of the Vehicle samples, and so only the other two TCEAL1 sgRNAs were included in our vehicle and docetaxel comparison (shown in Table 2). The results of this analysis has now been added to the relevant section in Results and Discussion, and as Appendix Table S4.

6. Is there any evidence to suggest that TCEAL1 depletion in Doc-resistant models would have a re-sensitising effect? Is there any clinical data to suggest that in Doc-resistant tumours that TCEAL1 levels are altered?

Response: In PubMed, there are 26 reports returned on searching for 'TCEAL1' and many of these reports are not cancer related. There are no publications related to treatment response or resistance. In addition, we could not find any data relating TCEAL1 to docetaxel resistance in prostate cancer from publically available datasets. Many of the online studies compare primary to metastatic tumours, but do not provide details about chemotherapy treatment that the patients have had. We did however find interesting information from two datasets (OncoPrint) where TCEAL1 was identified as significantly downregulated in studies comparing patients with metastatic disease to those with primary tumours. In a study by *Grasso et al.* TCEAL1 expression was 3.231 fold lower in metastases than the primary site (5). *Chandran et al.* saw a similar result with a fold change of -2.847 (6). We were however unable to identify which, if any, chemotherapy treatments may have been used to treat these patients and so it is difficult to draw any conclusions regarding a link between docetaxel treatment and TCEAL1 expression.

7. In Figure 3A, it is difficult to unpick the effect of TCEAL1 KD on Doc sensitivity because the effect of KD seems to be the same in both the presence and absence of Doc.

Response: We thank Reviewer 2 for this excellent comment. TCEAL1 was identified in our *in vivo* CRISPR screen as having an effect additional to docetaxel chemotherapy alone. The way our analysis was performed would not have taken into consideration the effects of silencing of individual specific genes. We focused on those changes that might sensitise cancer response to docetaxel treatment. The observed additional effect was also evident in our prostate cancer cell lines, including in PC3M cells as presented in Fig. 3A, which was meant to highlight the consequence of TCEAL1 knockdown in the presence of docetaxel, as well as to show that two individual siRNA had the same effect as the pooled siRNA. We appreciate the fact that presenting the data as one figure may be unclear, and therefore propose to split this figure into two parts, with the docetaxel treated samples remaining in the main figure (as updated Fig. 3A) and data from DMSO treated samples being presented in EV Fig. 3.

The impact of TCEAL1 knockdown in isolation seems to vary in different cells (Fig 2D, Fig 3C, new EV Fig 3). In future investigations, it would be interesting to study the molecular basis for such difference in TCEAL1 mediated effects on proliferation (and possibly other functional phenotypes). It is worth noting that the benign human prostate epithelial RWPE cell line does not seem to be affected by TCEAL1 silencing, suggesting a potential favourable therapeutic window for agents that targets TCEAL1 mediated function when used with taxane chemotherapy.

8. What is the expression of TCEAL1 in RWPE cells? In the FACS data, can the authors investigate if there is a synergistic or additive effect of KD and Doc treatments as it seems from eye-balling the data that the SubG1 level for the dual treatment/KD is the sum of KD and Doc alone.

Response: A Western blot showing the expression of TCEAL1 in prostate epithelial RWPE

compared to prostate cancer cell lines has now been added to EV Figure 3, and the results are described in the relevant section of the manuscript. The blot shows that TCEAL1 protein expression is almost undetectable in RWPE cells, although Figure EV3B shows an easily detectable expression at mRNA level, and that the KD is effective. The low level of TCEAL1 expression at the protein level in RWPE cells may contribute to the lack of anti-proliferative effects in RWPE cells following TCEAL1 KD (Fig 3C).

Using most methods, investigation to study whether two drugs (or in this case, a drug and a siRNA) are synergistic or additive requires a full range of concentrations of each treatment in combination with the other agent. In the absence of such data from formal combination index analysis, we hesitate to directly comment on presence of synergistic effects. Nonetheless, we observe that the percentage of subG1 cells with combined TCEAL1 knockdown (KD) and docetaxel treatment was averaged at 11.5% over the three biological repeats, while the individual subG1 fractions for TCEAL1 KD and docetaxel are 7.6% and 2.9% respectively. The fact that the effect of the combination is more than the simple addition of the individual treatments perhaps suggests that the combination is at least additive. While we have not sought to specifically demonstrate the presence of synergism, our data clearly showed that loss of TCEAL1 enhances the effect of docetaxel.

Reviewer #3 (Comments to the Authors (Required)):

Summary

The manuscript by Rushworth et al. centers around the results of an in vivo CRISPR knockout screen to identify candidate genes that act to limit the effects of (clinically relevant) docetaxel treatment on prostate cancer. The authors derived a murine cell line (SP1) from prostatic tumors in probasin-Cre Pten^{fl/fl} Spry2^{fl/+} mice that were subsequently modified to express CAS9 to enable CRISPR-based targeting of genes using a library of targeted guide RNAs. The modified SP1 cells were then implanted in the prostates of CD1 nude mice to form tumors and randomized into control and docetaxel treatment groups. Depleted guide RNAs were identified by bioinformatic analysis of sequencing reads of PCR-amplified regions of genomic DNA. The screen identified several candidate genes that the authors further examined by reducing their expression in cultured human prostate cancer cells using siRNA and assessing the resultant enhancement of the effects of docetaxel. The authors primarily focus on the top candidate TCEAL1 and further examined gene expression changes induced by decreasing its expression with siRNA in the absence and presence of docetaxel. The validation studies are relatively well described and provide reasonable evidence to support the authors' claims. The manuscript is well written, albeit generally descriptive in nature, providing sufficient evidence to implicate TCEAL1 and other potential candidate genes as modulators of docetaxel effects in prostate cancer.

Major critiques

1) In vivo versus in vitro effects. The authors make a major case for the importance of an in

in vivo screen to identify modulators of docetaxel effects on prostate cancer; however, all validation of the screen hits was performed in prostate cancer cell lines in culture. The authors do not provide any evidence that the results of the screen could be recapitulated *in vivo*. The importance of TCEAL1 as a modulator of docetaxel would be significantly strengthened by introducing the same sgRNAs targeting TCEAL1 (and relevant control sgRNAs) in SP1 cells and assessing the ability of knocking out this gene on docetaxel treatment of xenografted tumors. The authors suggest that reducing TCEAL1 gene expression enhances the effects of docetaxel, which should also equate to prolonged survival (or at least decreased tumor growth) of SP1-tumor-bearing mice.

Response: We thank Reviewer 3 for the positive and detailed comments above. As Reviewer 3 pointed out, we are reporting the first *in vivo* CRISPR screen in prostate cancer under treatment pressure, and highlighted candidate hits. We are keen to share our experience with the research community, including the potential pitfalls which we have commented on in the manuscript. Reviewer 3 is absolutely correct that *in vivo* validation of the role of TCEAL1 in tumour response to docetaxel treatment is required. The generation of cell models with suppressed TCEAL1 expression (preferably in an inducible manner) will be required for such *in vivo* experiments. In addition, the next stage of this project will also include mechanistic analysis of the basis of interaction TCEAL1 alone and in combination with docetaxel.

In addition, the authors provided survival data (Kaplan-Meier curves) only for the murine prostate cancer model from which the SP1 cells were derived. The authors state that weekly ultrasounds are used to assess tumor growth but they do not provide any data for the effects of docetaxel on SP1 tumor growth or mouse survival--data that would be more relevant to the genetic screen than the data from the original SP mouse model that is shown in Fig 1.

Response: We appreciate Reviewer 3's comment about data directly from the tumours during and at the conclusion of the *in vivo* screen. From previous *in vivo* experiments, including the pilot study shown in Figure 1, we found that docetaxel causes significant abdominal distension, which results in early removal of mice from the study. These *in vivo* pilot data are necessary to inform the proposed design of an *in vivo* screen. In the case of our screen experiment, the mice received only 3 docetaxel doses (4 days apart) before they had to be harvested due to abdominal distension. Based on our screen design, mice from the screen were harvested at the same time when clinical endpoint was observed in any of the experimental animals, making any survival data analysis less meaningful, particularly given the relatively short timeframe of the experiment. Figure 1I compares the final tumour volume of the orthografts from vehicle and docetaxel treated mice, showing that there is no significant difference. We have taken the opportunity to review our archival ultrasound images obtained as part of the *in vivo* screen experiment. In keeping with the final tumour volume, we did not observe significant difference in the orthograft volumes in mice treated with docetaxel and vehicle (see figure below summarising data from three serial ultrasound imaging).

Data from the pilot *in vivo* treatment experiment also confirmed the functional effects of docetaxel treatment. For the screen itself, to achieve sufficient material for sequencing analysis, the whole tumour (orthograft) from each mouse was ground up for DNA extraction. We also reasoned that given the nature of the screen at the level of individual cells, ‘global’ methods such as immunohistochemistry and bulk transcriptomic analysis would need to be performed on the same samples that were processed for sgRNA analysis, which is technically not feasible.

2) *The screen uses CRISPR/Cas9 genetic targeting, which would be expected to result in complete loss of gene expression whereas all validation was performed using siRNA, which dynamically alters expression (i.e. effects are time-dependent) and does not completely eliminate the expression of their target.*

Response: We thank Reviewer 3 for raising this interesting and pertinent point. While we agree with the reviewer that siRNA and CRISPR have fundamental differences in the manner in which gene expression is lowered, we think that the use of an orthogonal technique strengthens the validity of the top hits from the screen. Given the suboptimal performance of the screen, the fact that multiple targets could be validated using a different approach was reassuring.

Generating stable TCEAL1 CRISPR KO clones is certainly a relevant plan for future studies, but will take many weeks to generate in normal circumstances. In the current COVID-19 dominated situation, we are unable to both create the cell clones and satisfactorily repeat all relevant assays using the derived clones.

3) *Neither the data nor the analytical approach (computational tools) were provided or made available online, preventing deeper evaluation of the results. The author-provided link https://github.com/prepiscak/optichem_crispr is not valid (or the repository is not public). The data do not appear to have been deposited into public databases such as EBI or SRA.*

Response: We thank Reviewer 3 for this useful observation. The code repository on https://github.com/prepiscak/optichem_crispr is now public and visible to reviewers. CRISPR screen and RNA-seq data have been deposited in the ArrayExpress database at EMBL-EBI

(www.ebi.ac.uk/arrayexpress) under accession numbers E-MTAB-9482 and E-MTAB-9484, respectively. At the moment, data is kept private and reviewers can access private data using the following reviewer accounts –

Username: Reviewer_E-MTAB-9482 Password: zpcurjdc

Username: Reviewer_E-MTAB-9484 Password: gopYhkr

(https://www.ebi.ac.uk/arrayexpress/help/how_to_search_private_data.html). In addition, upon acceptance to publication, we will make the information available to the public in the near future.

Minor critiques

1) The potential role of androgen (dependence of tumor cells and deprivation therapy), a major factor in prostate cancer etiology and prognosis, is only peripherally described. More discussion is warranted.

Response: We thank Reviewer 3 for highlighting this important point. We have now added relevant information to the Discussion section. Docetaxel is often combined with ADT as an upfront treatment for metastatic prostate cancer, hence future studies to test the value of TCEAL1 mediated events as a potential target in the context of combined chemo-hormonal therapy is necessary. We also added new data in Fig. EV1B to demonstrate the expression of androgen receptor in the SP1 cells which confirmed its clinical relevance. Our previous research has revealed the importance of tumour *de novo* synthesis of androgens in driving androgen receptor function following androgen deprivation therapy (7).

2) The screen would produce hits for gene targets of the sgRNA library that are both under- and over-represented in the docetaxel-treated tumors compared to control yet only the under-represented genes are described. Similarly, genes essential for tumor growth would also be expected to be depleted in the untreated tumors compared to the pre-injection tumor cells. Since the essential genes are likely not a large proportion of the genes targeted by the library, it is not surprising that a summary statistic of sgRNA sequence abundance does not change between conditions (as depicted in EV2A). Examination of a subset of genes shown to be essential in other published studies could be useful.

Response: Reviewer 3 is absolutely correct that there were both under- and over-represented genes from the screen. We are particularly interested in genes that were under-represented in the screen as these genes may be required for cellular survival and/or proliferation, and could potentially be targeted for therapy. The screen did reveal gene hits that were significantly over-represented in the docetaxel treated tumours compared to the controls, and investigation to validate and study the underlying mechanism of these over-represented genes can be explored in a separate study in the future.

Not all genes labelled essential will actually be essential in SP1 cells as there is cell line variability. However, we expected to see around 10% of the ‘essential’ genes lost once the cells were transduced with the sgRNA library. The fact that we did not see this, and had only a relatively small percentage (~2%) of essential genes lost in the pre-injection cells compared

to the plasmid suggests that the screen performance was suboptimal. As we suggest in the Discussion, this may be related to inadequate coverage of sgRNA per gene as we used only library A.

We performed gene set enrichment analysis of essential gene sets as defined in CRISPRcleanR package (v0.5) (3) and found that these gene sets were not significantly depleted in the pre-injection cells compared to the plasmid. Contrary to what we expected, among all of the essential gene sets, only the Ribosomal Proteins gene set reached a significant adjusted p value.

Gene set	Normalised Enrichment Score	p value	Adjusted p value
RNA_polymerase	-1.27	0.1554	0.3627
Spliceosome	-0.82	0.7351	0.9762
Proteasome	-0.77	0.7825	0.9762
essential	-0.73	0.9762	0.9762
DNA_Replication	0.63	0.9279	0.9762
Ribosomal_Proteins	1.77	0.0068	0.0389

3) *By the authors' own admission, the statistical power of the screen was limited, possibly due to technical reasons. However, it is possible that the statistical power could be enhanced simply using alternative, more recent analytical tools such as JACKS for the CRISPR knockout analysis (<http://www.genome.org/cgi/doi/10.1101/gr.238923.118>).*

Response: We thank Reviewer 3 for this interesting suggestion. We appreciate that it may be useful to run the JACKS algorithm (or other algorithms and their combinations) to potentially identify additional hits from the screen. We therefore attempted to run the JACKS algorithm as suggested. However, the JACKS algorithm by default performs median normalisation (or uses control non-targeting sgRNAs) which is problematic due to the number of zero count sgRNAs in the screen. In the end, we were unsure how to interpret the results.

Regardless of the analytical tool used, we anticipate the key genes of interest, particularly Tceal1 as a potential target for validation, will remain valid and the main message of the report unchanged.

4) *Although "clinical end point" is used in the manuscript several times, it is never defined.*

Response: We apologise for this omission and have now added a description of the clinical endpoints to the Methods section. The clinical endpoints for this study were: tumour diameter greater than 1.2 cm; tumour invasion into other organs, including the bladder; and abdominal distension.

5) *The representation of the sgRNA library is shown as a boxplot distribution in Fig. 2A and independently for each gene in terms of $\log_2(\text{counts}+1)$. Some depiction of how the*

distribution of unique sgRNA abundances are maintained or altered in the different conditions is warranted. A waterfall plot of the abundances of each sgRNA in the plasmid library (in increasing or decreasing order) could serve as the baseline for comparison.

Response: We thank Reviewer 3 for this useful suggestion. Distribution of unique sgRNA abundances across different conditions were examined by plotting cumulative probability distributions as a function of normalised reads (see Fig. 1A below). In short, we observed more peaks at low or zero counts going from plasmid, cells, vehicle to docetaxel treatment, signifying increasing degree of depletion for subset of sgRNAs.

We appreciate that cumulative probability distributions is a perfectly valid and commonly used approach to assess distributions across conditions. We then followed Reviewer 3's suggestion and prepared a waterfall plot with ordered abundance using plasmid (see top left plot in Fig. 1B below) as a baseline for subsequent comparison. When we keep the plasmid order of individual sgRNAs in the other conditions as in top left panel of Fig. 1B, the plot becomes very difficult to follow (see Fig. 2 below).

To allow ease for cross-referencing, we have re-ordered on logFC for each of the comparisons and highlighted some of our key genes. We hope to highlight the relative abundance of the key genes of interest e.g. TCEAL1, and track their changes respective to plasmid across conditions (plots for cell, vehicle and docetaxel in Fig. 1B). Individual gRNAs for genes of interest are highlighted in colour to enable ease of cross-comparison. Figures 1A and 1B are now included in the manuscript as Figs. 2B and EV2B respectively.

Figure 1A. Cumulative probability distribution of sgRNAs in the plasmid, pre-injection cells, and vehicle and docetaxel treated tumour samples. Shift in the curves for vehicle and docetaxel treated tumour samples represents the depletion in a subset of sgRNAs after injection and after injection and docetaxel treatment, respectively. Distributions for each condition are averaged across replicates.

Figure 1B. Representation of whole genome sgRNA library in different conditions. Top left plot represents ranked sgRNA abundance in plasmid. Remaining plots represent ranked log2FC relative to the plasmid for different conditions. Log2FC were calculated using values averaged across condition.

Figure 2. Representation of whole genome sgRNA library in three different conditions with the order of sgRNA kept in order as it appeared in top left panel of Fig. 1B.

6) Y-axis labels "RQ" in Fig 4 (and related EV) are not defined.

Response: RQ has now been defined in the relevant figure legends as relative quantitation.

References

1. Patel R, *et al.* (2013) Sprouty2, PTEN, and PP2A interact to regulate prostate cancer progression. *J Clin Invest* 123(3):1157-1175.
2. Chen S, *et al.* (2015) Genome-wide CRISPR screen in a mouse model of tumor growth and metastasis. *Cell* 160(6):1246-1260.
3. Iorio F, *et al.* (2018) Unsupervised correction of gene-independent cell responses to CRISPR-Cas9 targeting. *BMC Genomics* 19(1):604.
4. Hodgkins A, *et al.* (2015) WGE: a CRISPR database for genome engineering. *Bioinformatics* 31(18):3078-3080.
5. Grasso CS, *et al.* (2012) The mutational landscape of lethal castration-resistant prostate cancer. *Nature* 487(7406):239-243.
6. Chandran UR, *et al.* (2007) Gene expression profiles of prostate cancer reveal involvement of multiple molecular pathways in the metastatic process. *BMC Cancer* 7:64.
7. Patel R, *et al.* (2018) Sprouty2 loss-induced IL6 drives castration-resistant prostate cancer through scavenger receptor B1. *EMBO Mol Med* 10(4).

September 17, 2020

RE: Life Science Alliance Manuscript #LSA-2020-00770-TR

Prof. Hing Y Leung
Beatson Institute for Cancer Research
Urology Group
Urology
Glasgow, Outside the United States or Canada G611BD
United Kingdom

Dear Dr. Leung,

Thank you for submitting your revised manuscript entitled "In vivo CRISPR/Cas9 knockout screen: TCEAL1 silencing enhances docetaxel efficacy in prostate cancer". We would be happy to publish your paper in Life Science Alliance pending final revisions necessary to meet our formatting guidelines.

Along with the points listed below, please also address the following:

please make sure the author list in our system and the manuscript match

-please separate the Results and Discussion into 2 separate sections - one Results section and another Discussion section

-please add an ORCID ID for the corresponding author-you should have received instructions on how to do so

-please use the [10 author names, et al.] format in your references (i.e. limit the author names to the first 10)

-please rename your Expanded View Figures as Supplementary Figures (e.g. update your figure legends and your figure callouts-(Fig. EV1A=Fig. S1A-page 5)

-please add a callout for Fig. 4C in your main manuscript text

-your Figure EV2 currently spans 2 pages; our figures must fit on one page; please update the figure to fit on one page or split your figure into 2 figures if you need more space

-please add scale bars to Figure 4A

A. FINAL FILES:

B. MANUSCRIPT ORGANIZATION AND FORMATTING:

Sincerely,

Shachi Bhatt, Ph.D.

Executive Editor
Life Science Alliance

Reviewer #1 (Comments to the Authors (Required)):

authors addressed the comments appropriately, both in tekst and by adding data (AR expression)

Reviewer #2 (Comments to the Authors (Required)):

The authors have responded very nicely to my initial comments and have adapted the text and data consistent with my queries. This is an excellent study- well done.

Reviewer #3 (Comments to the Authors (Required)):

The authors have reasonably addressed all of the concerns/critiques provided during the primary review process.

September 25, 2020

RE: Life Science Alliance Manuscript #LSA-2020-00770-TRR

Prof. Hing Y Leung
Beatson Institute for Cancer Research
Urology Group
Urology
Glasgow, Outside the United States or Canada G611BD
United Kingdom

Dear Dr. Leung,

Thank you for submitting your Research Article entitled "In vivo CRISPR/Cas9 knockout screen: TCEAL1 silencing enhances docetaxel efficacy in prostate cancer". It is a pleasure to let you know that your manuscript is now accepted for publication in Life Science Alliance. Congratulations on this interesting work.

DISTRIBUTION OF MATERIALS:

Again, congratulations on a very nice paper. I hope you found the review process to be constructive and are pleased with how the manuscript was handled editorially. We look forward to future exciting submissions from your lab.

Sincerely,

Shachi Bhatt, Ph.D.
Executive Editor
Life Science Alliance